

# Hypercarnivorous teeth and healed injuries to *Canis chihliensis* from Early Pleistocene Nihewan beds, China, support social hunting for ancestral wolves

Haowen Tong[1,2,3,*], Xi Chen[4], Bei Zhang[1,2,3], Bruce Rothschild[5], Stuart White[6], Mairin Balisi[7] and Xiaoming Wang[1,7,*]

[1] Key Laboratory of Vertebrate Evolution and Human Origins of Chinese Academy of Sciences, Institute of Vertebrate Paleontology and Paleoanthropology, Chinese Academy of Sciences, Beijing, China
[2] CAS Center for Excellence in Life and Paleoenvironment, Beijing, China
[3] University of Chinese Academy of Sciences, Beijing, China
[4] Nanjing Normal University, Nanjing, Jiangsu, China
[5] Department of Vertebrate Paleontology, Carnegie Museum of Natural History, Pittsburgh, PA, United States of America
[6] School of Dentistry, University of California, Los Angeles, Los Angeles, CA, United States of America
[7] Natural History Museum of Los Angeles County, Los Angeles, CA, United States of America
[*] These authors contributed equally to this work.

Corresponding authors
Haowen Tong, tong-haowen@ivpp.ac.cn
Xiaoming Wang, xwang@nhm.org

## ABSTRACT

Collaborative hunting by complex social groups is a hallmark of large dogs (Mammalia: Carnivora: Canidae), whose teeth also tend to be hypercarnivorous, specialized toward increased cutting edges for meat consumption and robust p4-m1 complex for cracking bone. The deep history of canid pack hunting is, however, obscure because behavioral evidence is rarely preserved in fossils. Dated to the Early Pleistocene (>1.2 Ma), *Canis chihliensis* from the Nihewan Basin of northern China is one of the earliest canines to feature a large body size and hypercarnivorous dentition. We present the first known record of dental infection in *C. chihliensis*, likely inflicted by processing hard food, such as bone. Another individual also suffered a displaced fracture of its tibia and, despite such an incapacitating injury, survived the trauma to heal. The long period required for healing the compound fracture is consistent with social hunting and family care (food-sharing) although alternative explanations exist. Comparison with abundant paleopathological records of the putatively pack-hunting Late Pleistocene dire wolf, *Canis dirus*, at the Rancho La Brea asphalt seeps in southern California, U.S.A., suggests similarity in feeding behavior and sociality between Chinese and American *Canis* across space and time. Pack hunting in *Canis* may be traced back to the Early Pleistocene, well before the appearance of modern wolves, but additional evidence is needed for confirmation.

## INTRODUCTION

Large, hypercarnivorous dogs (family Canidae)—such as gray wolves (*Canis lupus*), African hunting dogs (*Lycaon pictus*), and Asian dholes (*Cuon alpinus*)—are known to be highly social because of their need for collaborative hunting (*Van Valkenburgh, 1991*). In all three species, energetic requirements necessitate that they pursue prey species that are larger than themselves (*Carbone et al., 1999*). But, unlike their felid (cat family) counterparts, canids lack retractile claws and are usually unable to bring down their prey single-handedly (*Wang, Tedford & Antón, 2008*), making collaborative (pack) hunting a useful compensatory strategy. Despite the importance of pack hunting as a key biological indicator for social interactions, trophic relationship, and diets, however, fossil records rarely preserve direct information on behavior.

Discovery of an injured and healed skeleton and jaws of a large ancestral wolf, *Canis chihliensis*, from the Early Pleistocene hominin site of Nihewan Basin, northern China, is of interest in inferring their social behavior. Evidence of healing raises the possibility that individuals survived incapacitating injuries by sharing food with family members (*Palmqvist, Arribas & Martínez-Navarro, 1999*), a question to be explored in this article.

## MATERIALS & METHODS

The methods employed in this study include morphological observations, CT scanning, and X-ray examination. CT slicing intervals followed that of *Rothschild, Wang & Shoshani (1994)*. The osteological terms are from *Mescher (2018)*. The stages of fracture healing follow *Edge-Hughes & Nicholson (2007)*. Age determination follows *Sumner-Smith (1966)* for epiphyseal fusion and *Gipson, Ballard & Mech (2000)* for tooth wear. Body-mass estimates were calculated using regressions on canid femur shaft diameter by *Anyonge & Roman (2006)* and m1 length by *Van Valkenburgh (1990)*. Permission for excavation was granted by the State Administration of Cultural Heritage with a permit number of 2018-090.

**Locality and Fauna**. The present large sample of Early Pleistocene wolf, *Canis chihliensis*, comprises more than 200 specimens including excellently preserved pathological conditions. A left dentary (IVPP V17755.11), a right dentary (IVPP V17755.12), and a right tibia (IVPP V18139.20) of *Canis chihliensis* are all from the Shanshenmiaozui (SSMZ) Site in Nihewan Basin. *C. chihliensis* from SSMZ is dominated by older individuals as inferred from wear on teeth (*Chen, 2018*; *Chen & Tong, 2015*). The SSMZ locality (40°13′08″N, 114°39′54″E) lies at the southern bank of the Sangganhe River, and at the edge of the Haojiatai fluviolacustrine platform in Yangyuan County, Hebei Province (Fig. S1). The fossiliferous layer was dated to ca. 1.2 Ma by magnetostratigraphy and associated fauna (*Liu et al., 2016*; *Tong, Hu & Han, 2011*).

Canids are the most abundant carnivorans in the Early Pleistocene Nihewan Fauna (*Qiu, 2000*; *Teilhard de Chardin & Piveteau, 1930*), as also confirmed by our recent excavations at SSMZ (Fig. S2). The dominant taxon of the canid guild in the SSMZ Fauna is *Canis chihliensis* (*Tong, Hu & Han, 2011*; *Tong, Hu & Wang, 2012*). The mammalian fauna associated with *C. chihliensis* at the SSMZ site are as follows: *Lepus* sp., *Ochotona* sp.,

Pantherinae gen. et sp. indet., *Pachycrocuta* sp., *Mammuthus trogontherii*, *Coelodonta nihowanensis*, *Elasmotherium peii*, *Proboscidipparion* sp., *Equus sanmeniensis*, *Sus* sp., *Eucladoceros boulei*, *Spirocerus wongi*, *Bison palaeosinensis*, and *Gazella sinensis*. Our fieldwork between 2015–2018 recovered additional taxa, e.g., *Alactaga* sp. (represented by metacarpal), *Acinonyx* sp. (radius), *Panthera* sp. (partial mandible and manus bones), *Lynx* sp. (partial mandible with m1, mandible), *Paracamelus* sp. (partial metatarsal), *Pseudodama* sp. (partial antler and metacarpal), and *Gazella subgutturosa* (metatarsal) (*Tong & Chen, 2015*; *Tong, Chen & Zhang, 2017*; *Tong, Chen & Zhang, 2018*; *Tong, Hu & Han, 2011*; *Tong, Hu & Wang, 2012*; *Tong & Wang, 2014*; *Tong & Zhang, 2019*).

**Rancho La Brea *Canis dirus*.** The best records of paleopathology in extinct canids are from the world's largest collection of Late Pleistocene dire wolves, *Canis dirus*, from the Rancho La Brea asphalt seeps in Los Angeles, California, U.S.A. The Rancho La Brea paleopathology collection comprises about 3,200 specimens of dire wolves assembled from over 200,000 specimens representing a minimum of 3,500 individuals (dire wolves represent greater than 50% of all mammal specimens from Rancho La Brea) (*Shaw & Ware, 2018*). As the largest *Canis* that ever lived and presumably preferring larger prey, dire wolves are widely considered a social predator (*Anyonge & Roman, 2006*; *Carbone et al., 2009*; *Hemmer, 1978*; *Merriam, 1912*; *Stock, 1930*; *Van Valkenburgh & Hertel, 1998*; *Van Valkenburgh & Sacco, 2002*). The Rancho La Brea dire wolf collection preserves a range of pathological conditions throughout the skeleton (*Hartstone-Rose et al., 2015*; *Lawler, Widga & Smith, 2017*; *Moodie, 1918*; *Shaw & Howard, 2015*; *Stock, 1930*; *Ware, 2005*), with particularly debilitating examples offering evidence that strong social bonds existed to allow weakened or disabled individuals to survive for extended periods of time (*Shaw & Howard, 2015*; *Shaw & Ware, 2018*).

Focusing on *Canis dirus* from a single deposit (Pit 61/67) at Rancho La Brea, *Brown et al. (2017)* quantified patterns of traumatic pathology—injuries that likely resulted from hunting, including healed fractures and evidence of severe or chronic muscle strain as well as osteoarthritis—and predicted skull injuries to be common because of the probability of being kicked while chasing prey. Contrary to expectation, the cranium showed a low incidence of traumatic injury (1.6%) and the dentary even less so (0.18%) (*Brown et al., 2017*). This study, however, excluded dental injuries likely incurred from feeding—such as abscesses and alveolar resorption stemming from infection—which were also sustained by and preserved in *C. dirus* from Rancho La Brea. In the current study, we quantify these dental injuries, as well as traumatic damage to the dire wolf tibia, for comparison with dental and tibial injuries in *C. chihliensis*.

## TAXONOMIC AND PHYLOGENETIC REMARKS

As far as we are aware, there are few reports of debilitating injuries to large hypercarnivorous canines in the fossil record, including Early Pleistocene *Canis falconeri* from Venta Micena of Spain (*Palmqvist, Arribas & Martínez-Navarro, 1999*), *Cuon* from Late Pleistocene of Italy (*Iurino & Sardella, 2014*), and the latest Pleistocene occurrences of *Canis dirus* in the Rancho La Brea asphalt seeps (*Shaw & Howard, 2015*). This is despite a generally

excellent fossil record for large canids in the late Cenozoic because of canids' preference for mid-latitude open habitats, where terrestrial fossil records are best preserved and most extensively explored (*Tedford, Wang & Taylor, 2009*; *Wang, 1994*; *Wang, Tedford & Antón, 2008*; *Wang, Tedford & Taylor, 1999*).

The holotype of *Canis chihliensis* was originally described based on a maxillary fragment with P3-M2 from Feng-Wo at Huang-Lu village (Locality 64) in Huailai County, Hebei (Chihli) Province by *Zdansky (1924)*. *Teilhard de Chardin & Piveteau (1930)* referred additional specimens to this species from Nihewan Basin. *Rook (1994)* synonymized *C. chihliensis* with *C. antonii Zdansky, 1924*, but *Tedford, Wang & Taylor (2009)* returned to *C. chihliensis* by restricting the concept to large Nihewan *Canis*. The systematics of *C. chihliensis* from SSMZ has been treated by *Tong, Hu & Wang (2012)*.

*Rook (1994)* and *Sotnikova (2001)* referred the Pliocene-Early Pleistocene species *Canis falconeri* from Europe, *C. antonii* from Asia and *C. africanus* from Africa to the supraspecific group *Canis* (*Xenocyon*) ex gr. *falconeri*. All of them readily fall into the category of hypercarnivores based on dentition and *C. falconeri* has also been hypothesized to be a hypercarnivore similar to modern gray wolves (*Palmqvist, Arribas & Martínez-Navarro, 1999*). *Canis chihliensis* shares some similarities with *Sinicuon dubius* (*Tong, Hu & Wang, 2012*). Furthermore, *C. chihliensis* is among the largest *Canis* species of Eurasia in the Early Pleistocene.

## RESULTS

**Dental Fracture and Inflammations as Related to Bone-crushing and Hypercarnivory**. The left dentary (IVPP V17755.11) and right dentary (IVPP V17755.12) belong to the same individual. The left dentary (Figs. 1A–1D) has c, p1-3 and m2-3 intact, while the crown of p4, trigonid of m1, and mesial root of m1 are fractured and lost, apparently due to injuries suffered during life. Both root fragments of p4 are retained. On m1 only the talonid is preserved. Note on Fig. 1A that the alveolar bone in the region of the missing mesial root of m1 shows no residual socket, which indicates antemortem bone remodeling. This is consistent with the radiographic evidence of periapical bone resorption associated with the apices of the retained roots of p4 and the distal root of m1 (described below). There is also partial loss of the enamel on c and m1 and fracturing of the crowns of p2, p3, and root of m1. The pulp cavities of p4 and m1 are exposed. The dentin of all teeth is stained brown. All remaining cusps are moderately worn.

There are multiple fractures of the buccal and lingual cortical surfaces of the dentary, primarily in the regions of p2-p3, m1-m2, and the posterior surface of the mandibular ramus including the condylar process. All fractures appear to be postmortem as suggested by the absence of any repair.

There is loss of the cortical bone on the alveolar ridge in the regions of p3, p4, and m1. This was most likely caused by periodontitis *in vivo* although there may have also been some postmortem fracturing of the alveolar bone around m1.

The right dentary (Figs. 1E–1H) preserves i2-3, c, p1-4, and m1-2 *in situ*; the crown of m3 is missing, but one root tip remains deep in the alveolus. The crown of m1 is brownish

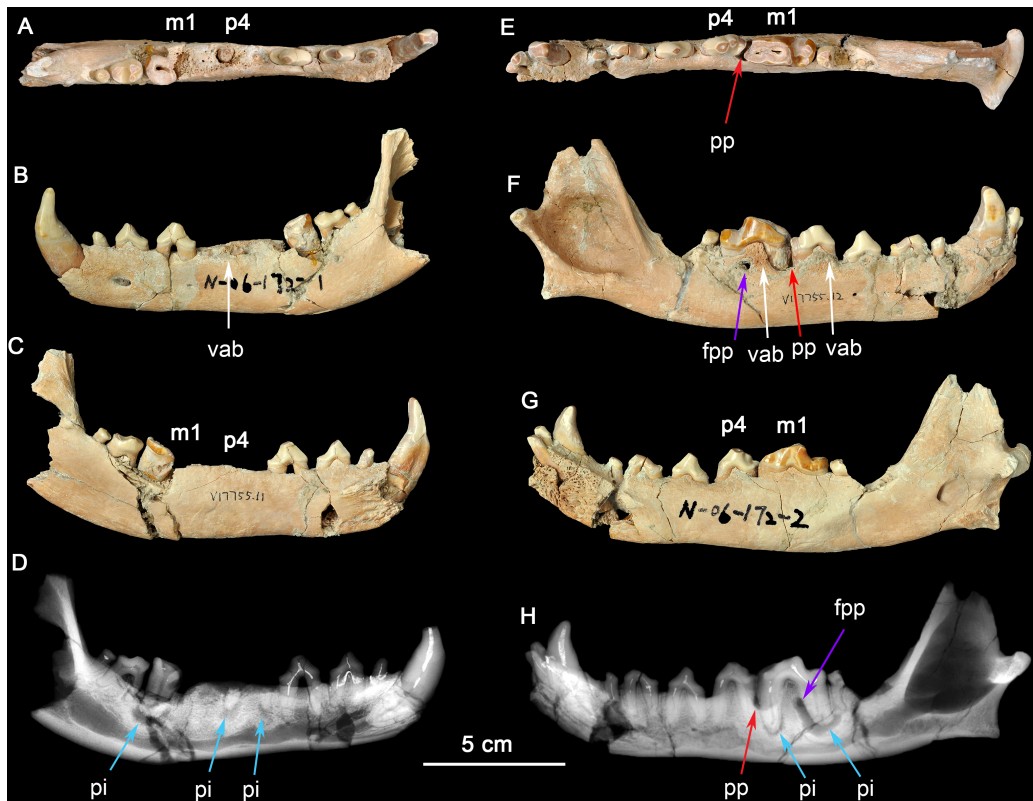

**Figure 1** **Two dentaries of the same individual of *Canis chihliensis*.** (A–D) Left dentary (IVPP V17755.11); (E–H) right dentary (IVPP V17755.12). (A, E) Occlusal views; (B, F) buccal views; (C, G) lingual views; (D, H) X-ray images. White arrows (labeled vab) indicate areas of increased vascularity of alveolar bone; red arrows (labeled pp) mark periodontal pocket, purple arrows (labeled fpp) indicate probable fistula from periodontal pocket, and blue arrows (labeled pi) mark periapical infections associated with exposed pulp chambers.

due to loss of most of the enamel cap, and with the pulp cavity exposed; m2 was broken during excavation; and other teeth are moderately worn. There are multiple fractures of the buccal and lingual cortical bone, predominantly in the regions of p1 and m2, that are postmortem defects.

The right dentary also suffered serious injury. The bone surrounding the m1 root is perforate on the buccal cortex (purple arrow, fpp, on Fig. 1H) by an apparent fistula and there is extensive loss of alveolar bone over the buccal aspect of the mesial root of m1 (red arrow, pp, on Fig. 1H). The buccal cortical surface is porous adjacent to p4 and m1 (white arrows, vab, on Fig. 1F). This is most likely the result of increased number and size of vascular canals associated with inflammation in this region.

**Radiographic observation.** The radiographic images of the right and left dentaries reveal periapical bone loss (rarefying osteitis) (blue arrows, pi, on Figs. 1D and 1H) associated with exposed pulp cavities, a periodontal pocket between the right p4 and m1 (red arrow, pp, on Fig. 1H), and an apparent fistula from the periodontal pocket to the surface (purple arrows, fpp, on Figs. 1F and Figs. 1H).

**Interpretation and implications for dental injury.** IVPP V17755 suffered from repeated dental injuries in similar locations on both left and right sides. Although both lever models and *in vivo* experimentation (*Ellis et al., 2008*) show that biting forces are greatest on the posterior-most molars, patterns of tooth wear suggest that the lower p4-m1 are used more frequently than more posterior molars (*Tseng & Wang, 2010*; *Wang, Tedford & Antón, 2008*; *Werdelin, 1989*), although in the case of the most hypercarnivorous canid, *Lycaon*, bone consumption may be at a more posterior location (*Van Valkenburgh, 1996*). Dental modifications for bone consumption in fossil borophagine canids are most apparent in the p4-m1 region, indicating that this was the location of most bone-cracking behavior (*Wang, Tedford & Taylor, 1999*). We interpret the loss of the left p4-m1 in IVPP V17755 as owing to bone-cracking—the p4 and m1 are the largest lower cheek teeth in *Canis* and their loss must have been inflicted by a strong biting force. Preservation of the roots of both the p4 and the m1 trigonid (Fig. 1D) suggests tooth fracture from a strong bite and/or encountering hard objects. The alveolar bone in the region of the missing m1 mesial root eventually healed, but the periapical infections associated with both retained root fragments of p4 and the distal root of m1 still show active lesions.

The need for bone-crushing in IVPP V17755 would have continued during and after the healing of the wounds on the left side. Accordingly, the right p4-m1 suffered excessive wear, likely to compensate for the loss of the same function on the left side. Again, we infer that the heavy wear is due to chewing on bones. The wear on the crown of m1 led to exposure of the pulp chamber through two pulp horns in the mesial cusp and directly to the periapical lesions (abscess) (blue arrows, pi, in Figs. 1D and 1H). This lesion grew sufficiently that it created a fistula to the buccal surface of the dentary to allow drainage of pus. It is also likely that excessive use on the right side led to bone splinters (shards, fragments) being imbedded into the gum tissue between p4 and m1, causing a periodontal pocket.

The above scenario suggests prolonged and possibly repeated injuries and infections, first to the left p4-m1 (possibly broken in a single bite), and then to the right jaw perhaps after the left side had partially healed. Such a scenario is consistent with a hypercarnivorous dentition in *C. chihliensis* frequently used for bone consumption, as also seen in Late Pleistocene European *Cuon* (*Iurino & Sardella, 2014*). Bone-crushing behavior in canids has been linked to collaborative hunting and competitive consumption of carcasses within the same family group of predators (*Wang, Tedford & Antón, 2008*; *Wang et al., 2018*). Such a behavior is especially prevalent among large, hypercarnivorous canids, and *Van Valkenburgh et al. (2019)* recently linked high tooth fractures in extant gray wolves to limited prey availability.

**Comparison to Rancho La Brea *Canis dirus*.** In Pit 61/67 alone, 35 dentaries of adult age (14 left, 21 right)—out of 64 pathological adult dentaries (25 left, 39 right; 55%) and 617 dentaries total (both pathological and non-pathological; 5.7%)—exhibit dental injuries similar to those in the Nihewan *C. chihliensis* dentaries examined in this current study (Fig. S3). Across Rancho La Brea deposits, abscesses and alveolar resorption likely due to infection were preserved in 43% (Pit 16) to 77% (Pit 3) of pathological dentaries (Fig. 2A). Most of the remaining pathological dentaries also preserved dental anomalies,

predominantly supernumerary teeth (particularly in the first and second premolars) or a missing lower first premolar (p1) and/or third molar (m3). Because both the p1 and m3 (*Balisi et al., 2018*; *Buchalczyk, Dynowski & Szteyn, 1981*; *Wang, 1994*) vary in their presence among canids, we excluded anomalies in these teeth from our comparison with Nihewan *C. chihliensis*. Across 200 *C. dirus* jaws (both left and right) bearing abscesses and alveolar infections, the lower first molar or carnassial showed the highest frequency of injury (87 total specimens with m1-associated injuries), likely inflicted by bone-crushing during the consumption of prey, followed by the second premolar (79 total specimens with p2-associated injuries), likely the result of biting and killing while chasing prey or in fighting with conspecifics or competitors of other species (Fig. 2B). The fourth premolar was the third most frequently injured tooth (57 specimens); often, it was injured in conjunction with the lower first molar (34 specimens), as in the case of *C. chihliensis*. As *C. dirus* is a predator widely recognized to have had a forceful bite capable of processing bone (*Anyonge & Baker, 2006*; *Brannick, Meachen & O'Keefe, 2015*; *Van Valkenburgh & Hertel, 1993*), the high frequency of injury in its p4-m1 complex—similar to that found in the specimens of *C. chihliensis* examined here—supports the inference that *C. chihliensis* also processed bone using p4 and m1.

**Tibia fracture**. A normal left tibia (IVPP V18139.21) and pathologic right tibia (IVPP V18139.20) of *Canis* are present in the collection from SSMZ. The pathologic tibia has healed fractures at the lower one-third of the shaft. Compared with the normal tibia on the left side (Fig. 3), the pathologic tibia is stouter; it is much broader distally, especially at the fracture site, and is shorter, the maximum length for the normal tibia being 181.6 mm, in contrast to the pathologic one at 166.5 mm (Table 1). In addition, the nutrient foramen is much more enlarged in the pathologic tibia. The partially healed bone has a rough and porous surface (callus).

The porous bone surface indicates that the periosteal vessels also took part in the repair of the fracture, which penetrated into the hard callus. Because the woven/primary bone is not replaced with secondary lamellar bone, this individual did not survive to the stage of lamellar bone formation, i.e., the fracture healing stage 6 by *Edge-Hughes & Nicholson (2007)*.

**Foreshortening of tibia.** The pathologic tibia has fused overlapping components with remodeling starting 4 cm from the proximal surface and extending throughout the length. Accentuation (irregularities) of the entheseal region at the lateral margin of the tibial plateau suggests increased stress at the proximal tibial-fibular joint. The tibia widens abnormally starting 6 cm distal to proximal surface, with concurrent alteration of surface color and texture, continuing on to the fused distal component of the tibial fracture, where surface filigree reaction (characteristic of infection) is more prominent. There are increased vascular markings at the junction of the proximal and middle third (related to current length) of the tibia. A shallow groove identifies the original demarcation of the fracture components now fused. The fibula was also fractured, and residual components are noted at the distal 6 cm. A linear defect is noted at the mid-portion of the tibia, slightly medial to the sagittal line. It appears to be perforated in a manner more suggestive of vasculature than of draining sinuses. It may be the residue of the fracture. If so, it would mean that the
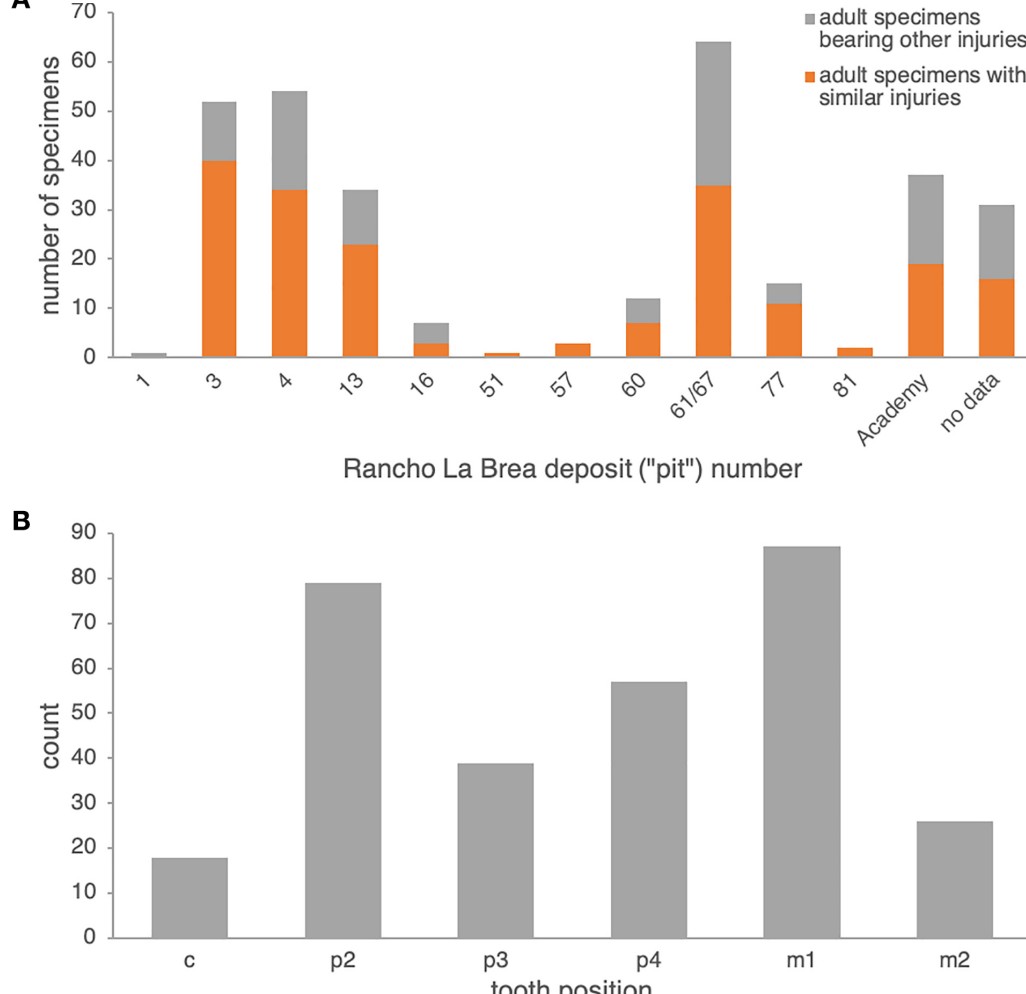

**Figure 2** **Frequencies of dental injury in the mandible of Rancho La Brea dire wolves, *C. dirus*.** (A) numbers of specimens of adult age bearing injuries similar to those in *C. chihliensis* (orange) compared with other dental injuries (gray). Most dental injuries in *C. dirus* involve abscesses and alveolar resorption stemming from infection. (B) categorization of dental injuries by tooth position. The m1 shows the highest frequency of infection or injury, followed by p2 and p4.

injury not only caused fracture, separation and overlap of components, but also caused a "splintering" or at least slight separation of the distal portion of the proximal component. Increased vascularity is noted 2 cm from the distal end of the tibia.

**X-ray examination.** Increased density of the medial tibial plateau is noted. If not related to an artifact (e.g., glued component), this is suggestive of a healed, minimally displaced fracture. There clearly is a displaced distal fracture, fused incompletely with overlap. The curvature of the distal portion of the proximal component suggests torsion of the components related to each other. Several layers of periosteal reaction are noted, with partial disruption of subjacent cortex. The distal fibula is fused to the tibia, with focal loss of margin definition. Irregular cavities are noted in the distal portion of the proximal
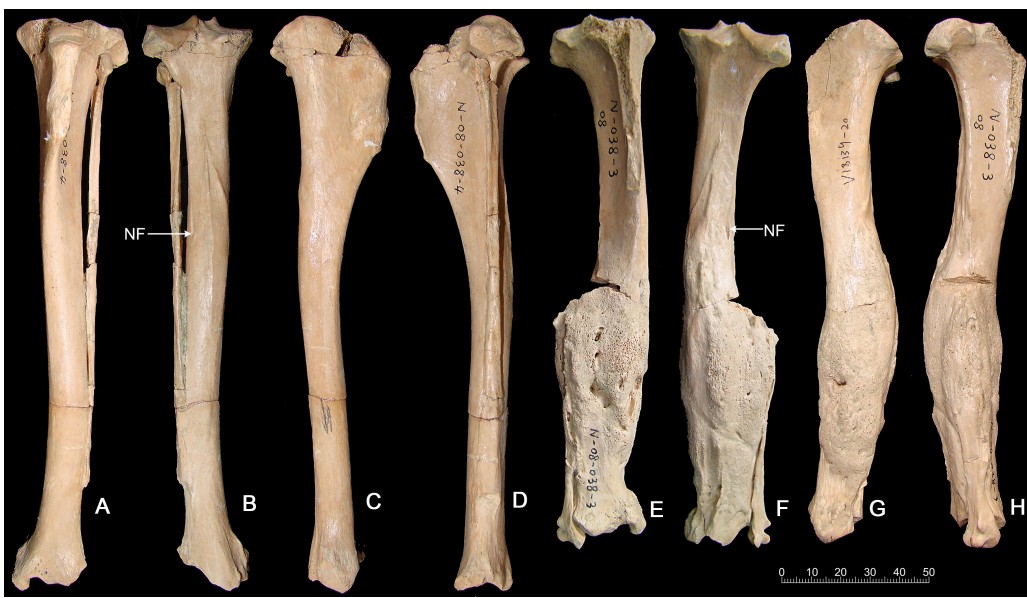

**Figure 3** **Tibias of the same individual of *Canis chihliensis* from SSMZ, Nihewan.** (A–D) Normal tibia of left side (IVPP V 18139.21). (E–H) Pathologic tibia of right side (IVPP V 18139.20). (A, E) Anterior views; (B, F) posterior views; (C, G) medial views; (D, H) lateral views. NF, nutrient foramen.

**Table 1** **Dimensional comparisons between the normal and pathologic tibiae of *C. chihliensis* (in: mm).**

| Dimensions | Normal (left) tibia (IVPP V 18139.21) | Pathologic (right) tibia (IVPP V 18139.20) |
| --- | --- | --- |
| Maximum length | 181.6 | 166.5 |
| Proximal DAP | 37.5 | >32.2 |
| Proximal DT | 36.5 | 35.8 |
| Distal DAP | 17.6 | >17.3 |
| Distal DT | 24.1 | 25.7 |
| Shaft DAP at nutrient foramen | 15.4 | 17.2 |
| Shaft DT at nutrient foramen | 13.2 | 14.8 |
| Shaft DAP at the fracture | – | 25.5 |
| Shaft DT at the fracture | – | 29.2 |

**Notes.**
Abbreviations: DAP, anteroposterior diameter; DT, transverse diameter.

component of the fracture and adjacent to the distal junction of the tibia and fibula. Both contain radio-dense material. This suggests that this was a compound fracture, with skin breach and exposure to environmental contamination. The fracture was incompletely stabilized during the healing process, with continued movement of the components.

**CT scan.** The CT images show clearly that it was a comminuted fracture, and all three pieces of the fractures are displaced, which resulted in the division of the medullary cavity into three chambers whose broken ends were enclosed by callus or woven bones (Figs. 4A–4D).

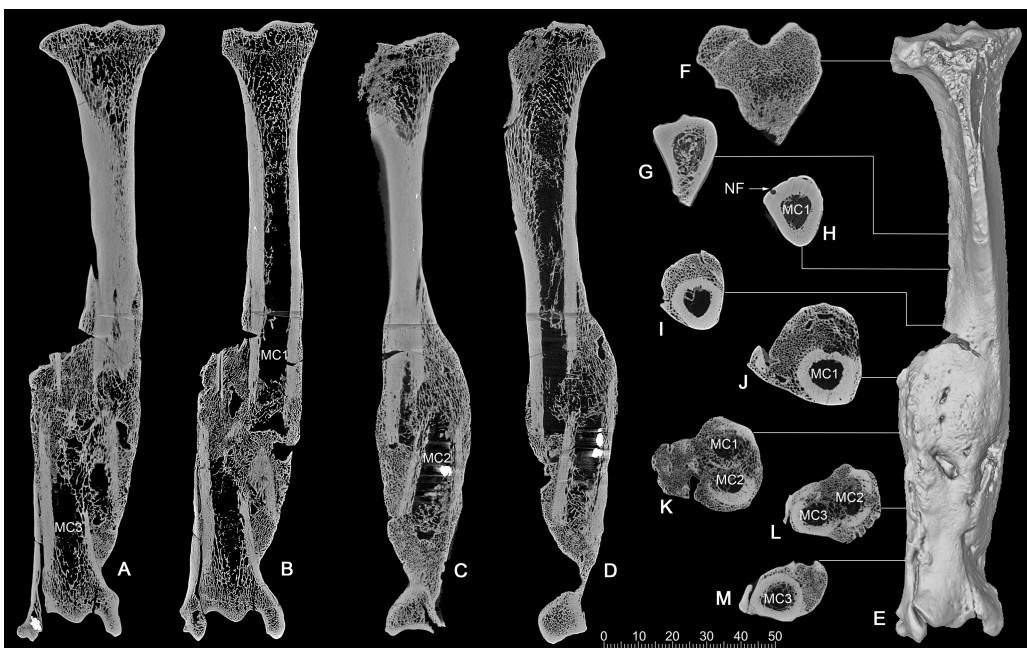

**Figure 4** **CT scan images of the pathologic right tibia of *Canis chihliensis* (V18139-20) from SSMZ, Ni-hewan.** (A–B) Anteroposterior longitudinal sections; (C–D) mediolateral longitudinal sections; (E) 3-D reconstruction of the pathologic tibia; (F–M) cross sections; (F–J) the upper part of the tibia; (K) the upper and middle parts of the fracture; (L) the middle and parts of the fracture; (M) lower part of the fracture, infection with subtle cortical loss. MC1-MC3, represent the medullar cavities of the three fractions of the fractured tibia; NP, nutrient foramen.

CT longitudinal sections slice 1 (Figs. 4A, 4B)—There is a focal area of trabecular loss just distal to the proximal epiphyseal plate. It is irregularly ellipsoid in shape and contains slightly thickened bone ''fragments'' of apparently increased density. Increased density is noted in the subsequent proximal fracture component. Periosteal reaction is noted with multiple focal areas of trabecular loss, bounded by sclerotic margins, characteristic of abscesses. There is massive loss of cortical bone in the region of fragment fusion. Fibular fusion with a distal radio-dense inclusion is noted. Presence of foreign bodies is consistent with the diagnosis of a compound fracture.

CT longitudinal sections slice 2 (Figs. 4C, 4D)—There is an area of increased density at the median tibial plateau noted on the X-ray. The CT shows this area to be separated by a fracture line from subjacent bone. The trabecular pattern is denser. The lateral portion of the proximal epiphyseal plate is partially preserved, in contrast to the medial portion, which cannot be distinguished from the epiphysis. This appears to be a non-displaced fracture through the epiphyseal plate, only affecting a portion of that plate.

There is a linear focal disruption (partially occluded at the surface) of the medial aspect at the midpoint of the current length and a U-shaped defect (also seen in CT slice 1) with thickened margins at the distal fifth. The latter could represent a draining abscess, although the former suggests the possibility of a penetrating injury. Radio-dense inclusions are noted, perhaps representing environmental exposure at time of injury. The surface imperfection

seen on the reconstructed tibial image (Fig. 4E) may be a CT averaging artifact. A series of 8 cross sections (Figs. 4F–4M) allows comparisons of healthy cancellous (F), healthy cortical (G-H), and injured and healed bones (I-M).

**Interpretation, comparison, and implications for limb injury.** That the injury, plus the subsequent infections, suffered by IVPP V18139 must have been devastating seems not in doubt. The displacement of the right hindlimb and the pain associated with a compound fracture with skin breach and exposure to environmental contamination all but rule out hunting activities. For modern domestic dogs of more than 1 year of age, fracture healing can take 7 weeks to 1 year (*Edge-Hughes & Nicholson, 2007*). Therefore, it is safe to assume that healing of the open fractures in IVPP V18139 without medical intervention (broken bones not re-aligned nor cast to immobilize wounds) would take a considerable amount of time, much longer than its metabolic reserve can sustain. Such a long-term survival by an injured wolf requiring a high degree of meat consumption thus suggests collaborative hunting and potentially family care.

In addition to abnormalities in the jaws and dentition, the Rancho La Brea dire wolf collection has numerous healed fractures in the limb bones (*Moodie, 1918*; *Shaw & Howard, 2015*; *Stock, 1930*; *Ware, 2005*). Again focusing on Pit 61/67, which has a minimum number of 371 dire wolf individuals, *Brown et al. (2017)* showed that frequencies of traumatic injury—including healed fractures—were higher than expected for most limb bones, especially the tibia. Surveying dire wolf tibiae across all Rancho La Brea deposits, we found 11 specimens (5 left, 6 right) of 251 total pathologic tibiae (4.38%) to have suffered an oblique fracture with foreshortening similar to that in IVPP V18139 (Fig. S4). In studies of modern Saskatchewan gray wolves and sympatric coyotes, such bone fractures—which likely resulted from conflicts with large prey—were found to be more common in wolves than in coyotes, a difference thought to result from wolves' tendency to prey on larger animals like moose (*Wobeser, 1992*). Similarly, Rancho La Brea preserves no fractured and healed tibiae belonging to the coyote—which is also found abundantly in the Pleistocene to Holocene-age asphalt seeps—though this lack may be confounded by a coyote sample size an order of magnitude smaller than that of the dire wolf.

## DISCUSSIONS

Life is not easy for large predators. In modern canids, hypercarnivory is almost always associated with social hunting, such as in the gray wolves (*Canis lupus*), African hunting dogs (*Lycaon pictus*), and Asiatic dholes (*Cuon alpinus*). Of these, the latter two most hypercarnivorous species almost invariably hunt cooperatively, whereas gray wolves regularly, but not exclusively, hunt together for large prey (*Macdonald, 1983*). Group hunting by these highly social canids offers apparent advantages that are otherwise unavailable to individual hunters, such as the ability to bring down prey much larger than the predators themselves, plus coordinated attacks that seal off escape routes as well as relaying strategies that lessen the burden of individual hunters. These strategies are especially critical to canids because, unlike felids, canids never evolved fully retractile claws that are effective weapons for grappling with and subduing prey (*Wang, 1993*). Therefore,

for canids, group hunting is not optional, as it is for large cats (only the lions are social hunters, as are occasionally the cheetahs), once canids have crossed the critical body mass threshold of about 21 kg above which energetic costs necessitate feeding on large prey (*Carbone et al., 1999*). For canids, it is possible that this body size threshold may even be substantially lowered as in the case of the Asiatic dholes (10–13 kg) that have the most extremely hypercarnivorous dentitions among living canids (*Cohen, 1978*). The Nihewan *Canis chihliensis* is larger than the dholes (13.7–16.8 kg based on femur shaft diameter; ~21.2 kg based on the mean of m1 length).

Social hunting is characteristic of large canids, hyaenids, and some felids, and depending on how such behavior is described, may even be quite common in carnivorans (*Bailey, Myatt & Wilson, 2013*). Such behavior has important implications not only in the social organizations of large carnivorans but also in their trophic relationships and diet. Among large, hypercarnivorous living canids, the gray wolf (*Canis lupus*) is the best studied in its pack hunting behavior. The basic social unit is the mated pair; prey size is a factor in pack sizes, which range from a few up to 20 individuals, with the largest packs preying on bison and moose and smaller packs preying on deer (*Mech & Boitani, 2003*). Social hunting, however, may not always be the most efficient in terms of food intake per wolf because the packs must share their proceeds (*Thurber & Peterson, 1993*). The formation of packs, therefore, offers the opportunity to kill prey too large to tackle by one individual alone, as well as the opportunity both to better defend kills against carcass theft and to steal carcasses from larger predators (*Carbone, Du Toit & Gordon, 1997*; *Eaton, 1979*; *Van Valkenburgh, 2001*; *Vucetich, Peterson & Waite, 2004*).

It has been long known that large *Canis* from the Nihewan Basin includes individuals with highly trenchant lower molars (*Teilhard de Chardin & Piveteau, 1930*). Hypercarnivorous characteristics (dominance of cutting edge of m1 trigonid and enlargement of hypoconid at the expense of entoconid, along with reductions of posterior molars) in *C. chihliensis* are variable (*Tong, Hu & Wang, 2012*) but strongly converge on the morphology of living African hunting dogs and Asiatic dholes (Fig. 5). Such a dental morphology is commonly associated with emphasis in slicing meat using the sharp carnassial blades. Trenchant molars thus correlate well with hypercarnivory (*Crusafont-Pairó & Truyols-Santonja, 1956*), i.e., tendency to consume meat exclusively, which also drives the evolution of larger body size as a macroevolutionary ratchet (*Van Valkenburgh, Wang & Damuth, 2004*).

Wolves have a dangerous life as long-distance pursuit predators. The traumas and infections inflicted on *Canis chihliensis* likely are related to hunting behavior, feeding strategies, and predator–prey interactions, as have also been suggested for other extinct carnivores (*Shaw & Ware, 2018*). Healing from such devastating injuries is also a testimony to its survival for long periods of time during which the ability to hunt must have been seriously limited or nonexistent, suggesting that assisted living was a possibility. Debilitating bone diseases in the Pleistocene apex predator *Smilodon*, which were even more hypercarnivorous than canids, have also been used to argue for social or gregarious behaviors (*Akersten, 1985*; *Heald, 1989*; *Shaw, 1992a*; *Shaw, 1992b*; *Van Valkenburgh, 2009*; *Van Valkenburgh & Sacco, 2002*) although the pathology-sociality link has been challenged (*McCall, Naples & Martin, 2003*). *Schleidt & Shalter (2004)* also noted that social predators

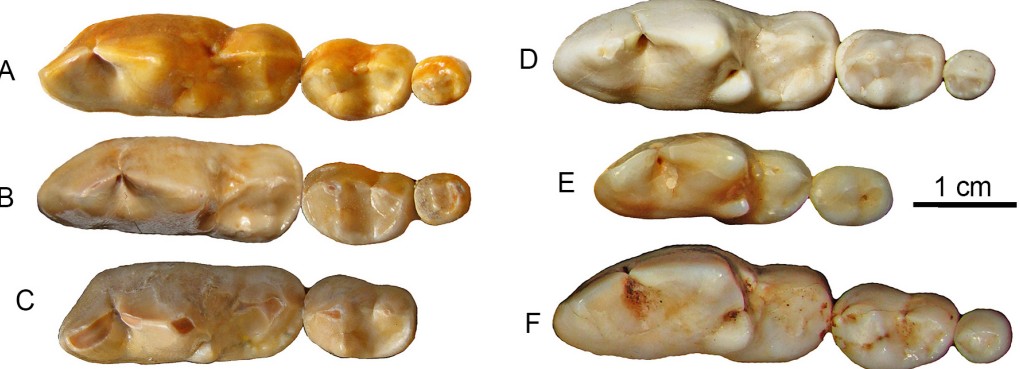

**Figure 5  Lower molars from SSMZ as compared to living hypercarnivorous taxa.** Occlusal views of lower molars, m1-3, of *Canis chihliensis* (A–C) from SSMZ in Nihewan, as compared with those of *C. lupus* (D), *Cuon alpinus* (E) and *Lycaon pictus* (F). (A) right m1-3 (IVPP V17755.6); (B) right m1-3 (IVPP V17755.4); (C) left (inverted) m1-2 (IVPP V17755.5); (D) right m1-3 (IOZ no number, extant, China); (E) right m1-2 (IOZ 26747, extant, China); (F) right m1-3 (T.M. No. 5560 and BPI/C 223, extant, South Africa). Modified from *Tong, Hu & Wang (2012)*.

should have more healed injuries than solitary predators. Often infirm animals are allowed to feed on group kills, as observed in spotted hyaenas and African wild dogs.

Whereas sociality in sabertooth cats has been questioned given its rarity among extant large felids, all of which are capable of killing on their own, pack hunting in dog-like carnivorans (wolves, hunting dogs, dholes, hyenas) is the dominant mode of predation and may partly be driven by the necessity of overcoming larger prey (*Mech & Boitani, 2003*). Dental morphology and pathology in our Nihewan *Canis chihliensis* strongly suggest processing of hard food (bone cracking), which is commonly associated with hypercarnivory and pack hunting in large canids. While herbivores, too, suffer from crippling injuries, comparisons to herbivores are irrelevant in this case because injured herbivores can continue eating plant matter, foraging on food items that do not move, while recovering from injuries. However, critical carnivore injuries, such as to the running hindlimbs, blunt active predators' ability to hunt and chase animal prey. Although the massive, healed tibial fracture may not be a definitive indication of social care, a predator's recovery from such a devastating injury is suggestive of food provisioning that only social groups can offer. This has been similarly proposed from an Early Pleistocene Spanish record of *C. falconeri* (*Palmqvist, Arribas & Martínez-Navarro, 1999*), although temporary shift to a more omnivorous diet is also possible. With this new record from Nihewan, we extend the history of *Canis* sociality to the Early Pleistocene, and likely to the Pliocene as well if the even larger *Canis antonii* from Fugu area in Shanxi Province is taken into consideration (*Tedford, Wang & Taylor, 2009*: Appendix I).

Arguably the most definitive (though still correlative) pathological evidence to support sociality in *Canis chihliensis* would be a significant prevalence of similar injuries not only in the extinct *Canis dirus* but in the three extant hypercarnivorous canines whose pack-hunting behavior can be observed directly, in contrast to a low prevalence of similar
injuries in non-pack-hunting carnivoran species. However, one common challenge in predator paleopathology is the lack of sufficient samples of large-predator post-crania relative to crania in museum collections of living mammals. Survival with just the leg or just the dental damage does have isolated representation, but not the combination. Museum records of similar injuries and survivals undoubtedly exist for non-bone-crunching and non-social species as well (but published documentation is often lacking) and a definitive inference is not possible without more detailed records, both extant and extinct. This limitation—and the corresponding lack of published systematic pathological surveys across large sample sizes within and among extant species—prevents statistically robust inferences of injury prevalence in extant wild animals. When isolated cases are available, lack of field documentation on behaviors related to pathological specimens also hampers interpretations. Such deficiencies make it difficult to ground-truth inferences of extinct behaviors based on extant relatives, even where large samples of extinct predators are available (*Brown et al., 2017*). While such a systematic comparative survey exceeds the scope of the current paper, future studies that calculate injury prevalence across large museum and zoo collections of extant species of known behavior (e.g., *Rothschild, Rothschild & Woods, 1998*) would bolster inferences of extinct behavior based on skeletal injuries.

As knowledge of the fossil history of hypercarnivorous canids in the Plio-Pleistocene of Eurasia increases, more complexity than has been previously assumed is now emerging, both in its chronology and its morphologic diversity. Recent molecular studies placed *Cuon* and *Lycaon*, two of the most hypercarnivorous living canids, near the base of the *Canis* clade (*Chavez et al., 2019*; *Koepfli et al., 2015*; *Lindblad-Toh et al., 2005*), in contrast to morphological analysis suggesting that hypercarnivorous forms are at the terminal end of the canine phylogeny (*Tedford, Taylor & Wang, 1995*; *Tedford, Wang & Taylor, 2009*). If the molecular relationship is correct, then records of *Cuon* and *Lycaon* are expected to be at least as old, if not older, than that of many species of *Canis*. This new record pushes back the first occurrence of pack hunting likely accompanied by social care by about 1.7 million years to when early *Homo erectus* was first recorded in Asia (*Ao et al., 2013*; *Zhu et al., 2004*). This record is important because it coincides with the initial diversification of the large canids (such as *Canis* and *Lycaon*), also known as the Wolf Event in Eurasia (*Azzaroli, 1983*; *Sardella & Palombo, 2007*), and *Lycaon*'s arrival in Africa (*Hartstone-Rose et al., 2010*).

Although records of early wolves have been pushed back slightly (*Martínez-Navarro, Belmaker & Bar-Yosef, 2009*; *Rook & Martínez-Navarro, 2010*; *Sardella & Palombo, 2007*), the wolf event is essentially confined to the Early Pleistocene, i.e., Late Pliocene before recent redefinition (*Gibbard et al., 2010*). A recent new Tibetan record in the Middle Pliocene, *Sinicuon* cf. *S. dubius*, seems to suggest that hypercarnivorous canines may have predated the genus *Canis* (*Wang, Li & Xie, 2014*). Whatever the detailed relationships of these records, it seems clear that hyper-predators, such as large wolves and hunting dogs, were associated with the increasingly open habitats in Eurasia during the onset of the Pleistocene. In this background of large-canine radiation at the beginning of the Ice Age, our new record of a pathological wolf from the Early Pleistocene of Nihewan hints at pack

hunting as a major step toward social collaboration while procuring food and, as such, signals a major step in the evolution of large canids.

## CONCLUSIONS

We document dental injuries and infections and a healed tibia fracture in *Canis chihliensis* from the Early Pleistocene (>1.2 Ma) Nihewan Basin of northern China. This early species of wolf-like *Canis* signals the evolution of large body size and hypercarnivorous dentition in the genus. The dental injuries and infections likely occurred while processing hard food, such as bones, whereas the tibia fractures would have severely limited locomotion during recuperation. Dental injuries and healing of compound fracture supports social hunting and family care (food-sharing) although alternative explanations exist because similar injuries likely appear in non-bone crunching and non-social species as well. Comparisons with abundant paleopathological records of the putatively pack-hunting Late Pleistocene dire wolf, *Canis dirus*, at Rancho La Brea in southern California demonstrates similarity in feeding behavior and sociality between Chinese and American *Canis* across space and time.

**Institution and Locality Abbreviations**

| | |
|---|---|
| **HPICR** | Hebei Province Institute of Cultural Relics |
| **IVPP** | Institute of Vertebrate Paleontology and Paleoanthropology |
| **MNHN** | Muséum national d'Histoire naturelle |
| **NM** | Nihewan Museum |
| **NNNRM** | Nihewan National Nature Reserve Management |
| **SSMZ** | Shanshenmiaozui |
| **TNHM** | Tianjin Natural History Museum |
| **V** | Prefix in the catalog numbers for vertebrate fossils in IVPP |

**Morphological Abbreviations**

| | |
|---|---|
| **DAP** | anteroposterior diameter |
| **DT** | transverse diameter |
| **MC** | medullar cavity |
| **NF** | nutrient foramen. |

## ACKNOWLEDGEMENTS

The authors wish to express their thanks to the following people and organizations for their help: Han F, Sun B. Y., Lü D., Sun J. J., Xu Z. J., Qiu Z. W., Wang Q. Y., Sun B. H., Hu N., Liu X. T. & Yin C. for participating the fieldwork; Xie F. of HPICR, Zhao W. J. of NNNRM and Hou W. Y. of NM for help during excavations; Qiu Z. X., Wei Q. for sharing bibliographies and/or for fruitful discussions; Hou Y. M. for CT scanning; F. Heald and C. Shaw for initial diagnosis and assembly of the Rancho La Brea pathology collection; A. Farrell and G. Takeuchi for Rancho La Brea collections access; B. Van Valkenburgh for thoughtful critique. We are grateful to Julie Meachen, Josh Samuels, and an anonymous

reviewer (who kindly reviewed our paper twice) for their critical reviews and comments, and editor Virginia Abdala for her editorial suggestions.

### Funding

This work was supported by the following grants: The Strategic Priority Research Program of Chinese Academy of Sciences (Grant No. XDB26000000); The National Natural Science Foundation of China (Grant No. 41572003); The Special Basic Research Project (Grant No: 2014FY110300) of MST of China. The funders had no role in study design, data collection and analysis, decision to publish, or preparation of the manuscript.

### Grant Disclosures

The following grant information was disclosed by the authors:
The Strategic Priority Research Program of Chinese Academy of Sciences: XDB26000000.
The National Natural Science Foundation of China: 41572003.
MST of China: 2014FY110300.

### Competing Interests

The authors declare there are no competing interests.

### Author Contributions

- Haowen Tong conceived and designed the experiments, performed the experiments, analyzed the data, prepared figures and/or tables, authored or reviewed drafts of the paper, and approved the final draft.
- Xi Chen performed the experiments, prepared figures and/or tables, field works and taphonomical analysis, and approved the final draft.
- Bei Zhang performed the experiments, prepared figures and/or tables, field works, and approved the final draft.
- Bruce Rothschild analyzed the data, authored or reviewed drafts of the paper, and approved the final draft.
- Stuart White and Mairin Balisi analyzed the data, prepared figures and/or tables, authored or reviewed drafts of the paper, and approved the final draft.
- Xiaoming Wang conceived and designed the experiments, analyzed the data, prepared figures and/or tables, authored or reviewed drafts of the paper, and approved the final draft.

### Field Study Permissions

The following information was supplied relating to field study approvals (i.e., approving body and any reference numbers):
State Administration of Cultural Heritage approved the study (2018-090).

### Data Availability

The raw data are available in Figs. 1–5, Table 1, and the Supplemental File.

All specimens described in this paper are deposited in the Institute of Vertebrate Paleontology and Paleoanthropology (IVPP), Chinese Academy of Sciences. The IVPP catalog system does not have accession numbers, they only use catalog numbers.

## Supplemental Information

Supplemental information for this article can be found online at http://dx.doi.org/10.7717/peerj.9858#supplemental-information.

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
