# Peer review of "Hypercarnivorous teeth and healed injuries to Canis chihliensis from Early Pleistocene Nihewan beds, China, support social hunting for ancestral wolves"

_PeerJ, doi:10.7717/peerj.9858_

## Round 0.1 · original submission · Major Revisions

I received three reviews of your manuscript. Two of them, #1 and #3, are mostly positive. Reviewer #1 makes a warning related to the phylogenetic relationships of Canis dirus and the modern canis. Reviewer #3 suggests a reorganization of your work. Both suggest several additional papers that should be included in your study.

Reviewer #2 considers that you go far beyond your data allow you to infer, and I concur. You claim that the pathologies you find in your sample indicate processing hard food, social hunting, and family care. As far as I can see, you are making an inference over another inference, and this is excessive. Moreover, the reviewer stressed that it is easy to find similar pathologies in both fossil and modern collections in taxa that neither crunch bones nor are communal hunters. What about these dental pathologies being common in many species of animals (including herbivores) in individuals that live to ancient ages? I find these criticisms highly compelling.

I would like you to write your inferences in a much less restrictive way, starting from the title. Of course, you can suggest social hunting and family care (processing hard food sounds much more probably), but suggesting is different to stress that you have extended the history of Canis sociality to the early Pleistocene.

·

Basic reporting

I though everything looked good

Experimental design

I especially liked the comparison to the La Brea Tar Pits to better understand the ecology of this extinct canid.

Validity of the findings

No comment

Additional comments

I reviewed this manuscript previously, and I thought it was great then. I don't quite understand all of the issues the other reviewers had. I also wasn't able to upload my minor changes in my previous review, so I am uploading them here.

Reviewer 2 ·

Basic reporting

Very professional article (with a clearly disprovable central premise)

Experimental design

The flawed central premise (see below) calls into question the experimental design: the "question [(i.e., is this taxon a durophage and group hunter) is NOT] relevant [or] meaningful" given that it can't be answered from these fossils.

Furthermore, because of this flawed logic, the "investigation [is NOT] performed to a high technical ... standard".

The methods are detailed and replicable.

Validity of the findings

Absolutely NOT! See below.

Additional comments

This is a well written paper by Tong, Wang and colleagues about some interesting pathological specimens of an ancient large dog. Unfortunately, the central claim of their paper is entirely indefensible: they claim that these extensive pathologies indicate not only the “processing hard food, such as bone” but also that they are evidence of “social hunting and family care (food-sharing)”. The claims are so strong that they lead to the conclusion that “Pack hunting in Canis can thus be traced back to the early Pleistocene, well before the appearance of modern wolves.” Parsimoniously, we would assume that this taxon is a hypercarnivorous pack hunter that likely ate bones – like most large canids. However, it is completely impossible to infer this merely from these pathologies – and especially offensive for them to do so to such an extent as this definitive claim. As the authors must know, it is quite easy to find similar pathologies in both fossil and modern collections in taxa that neither crunch bones nor are communal hunters. Would the authors acknowledge that these type of dental pathologies are common in many species of animals (including herbivores) in individuals that live to very old ages? Would they also confirm that they have seen similarly healed fractures in taxa that couldn’t possibly food provision (e.g., herbivores or solitary carnivores or rodents, etc.)? These are certainly rare, but still fairly easy to find in collections. If these comparative specimens exist, then it is absolutely false to claim – especially with the outlandish certitude of the language in this paper – that they MUST equate to bone crunching and food provisioning. It is hard to review the details of this paper when its central claim is so disprovable, though there is clearly good paleontological description in this piece.

·

Basic reporting

The Methods indicate "The osteological terms are from Mescher (2018)", but that source is not included in the references.

The Results starts with a section entitled "Taxonomic and Phylogenetic Remarks", lines 127-148, which really represents background information on Canis chihliensis and other similar canids. There is no new analysis presented in that section, but rather a review of the current understanding of large, hypercarnivorous canines. I would recommend relocating this section in its entirety to be part of the (rather short) introduction.

Figure 1 includes a number of color coded arrows indicating key pieces of information. Those may be difficult to interpret for color-blind individuals, and I would recommend adding some simple labels (i.e. abbreviations like those in Figures 3 and 4) to make this figure accessible to all readers.

Experimental design

No Comment

Validity of the findings

No Comment

Additional comments

Overall, I enjoyed reading this manuscript and found it to be very well-written. In addition to my comments related to basic reporting, I have added some additional comments to an annotated pdf file. Most of my suggestions are relatively minor and should be very easily addressed.

---

## Round 0.2 · Minor Revisions

It seems that we are close to the final review step. However, I agree with our second reviewer that you should change the word "suggest" in the title by "support." Please make a clear caveat in the body of the text that these types of injuries appear in non-bone crunching and non-social species as well. If you agree to consider these two aspects, I think that we can go ahead. Thank you.

Reviewer 2 ·

Basic reporting

See below

Experimental design

See below

Validity of the findings

See below

Additional comments

Again, this is a well written paper with a demonstrably false premise! Yes, this taxon was almost certainly a bone cruncher and pack predator – as the authors point out, this is the case for all modern large dogs today and is therefore the parsimonious assumption. However, the authors disingenuously claim that they are making this argument on the totality of this parsimony when THEIR TITLE makes the direct claim that they are making this assertion based on “teeth and healed injuries”. Their arguments may be “more subtle”, but their central assertion is loud and clear!

They assert that I was “offended by [their] linkage between a broken tibia and social hunting”, but this is incorrect: I’m not offended by this demonstrably wrong linkage (many of us are wrong all of the time!) or even their adherence to this demonstrably wrong assertion (though I don’t understand why they are so dug in about this). Rather, I am offended by their lack of acknowledgement that they have seen these types of injuries in other non-social hunters. I acknowledge that herbivores might handle this differently, but surely they have seen these injuries in other non-social carnivores. They are rather common in some cats, and I have seen them in ursids and mustelids too. Surely these excellent scientists have seen them in their combined decades of experience, right? I haven’t had the pleasure of visiting the LACM collection, but I am confident that such healed bones must exist in there, as they do in the AMNH, USNM and museums in Europe. (I suggest a visit to the leopard collection as a likely place to find some. If there are more than 20 wild individuals in that collection, then I will buy all of the authors dinner if there isn’t at least one leopard with similar dental ailments and SOME evidence of healed injury.)

The authors do make an exceptional observation in their rebuttal: I did not cite these beyond personal experience, and, come to think of it, I am unaware of any publication documenting these type of injuries. I am sure that Meachen, Sammuels of Van Valkenburgh could direct us to that literature as they are more knowledgeable about pathology. If this is indeed a deficit in the literature, then someone should document it!

As the authors note, if you knew nothing about these dogs other than that they were large dogs, then their assertion that they were likely bone crunchers and social would be the parsimonious one. Unfortunately, that is not the central claim of this paper.

With all of that said, this is a well written piece of work from a group of respected colleagues. If the editors agree that it is up to the journal’s standard, then I will withdraw my objection if the authors change ONE word in the title: if the word “suggest” is replaced with “support”, then I think this is accurate. I would BEG the authors to add an acknowledgement in the body of the text that these types of injuries are found in non-bone crunching and non-social species as well. I assume that they know this from their own work in collections and apologize that I do not know of a good formal citation for that. If I were the editor of this piece, I would insist on these two addenda.

---

## Round 0.3 · accepted · Accept

Thank you for your careful consideration of our reviewer suggestions. We are ready to move on. Nice work.